# Superconducting Stiffness and Coherence Length of FeSe$_{0.5}$Te$_{0.5}$ Measured in a Zero-Applied Field

**Amotz Peri \*, Itay Mangel and Amit Keren** 

Department of Physics, Technion-Israel Institute of Technology, Haifa 3200003, Israel
\* Correspondence: amotzpery@gmail.com

**Abstract:** Superconducting stiffness $\rho_s$ and coherence length $\xi$ are usually determined by measuring the penetration depth $\lambda$ of a magnetic field and the upper critical field $H_{c2}$ of a superconductor (SC), respectively. However, in magnetic SC, which is iron-based, this could lead to erroneous results, since the internal field could be very different from the applied one. To overcome this problem in Fe$_{1+y}$Se$_x$Te$_{1-x}$ with $x \sim 0.5$ and $y \sim 0$ (FST), we measured both quantities with the Stiffnessometer technique. In this technique, one applies a rotor-free vector potential **A** to a superconducting ring and measures the current density **j** via the ring's magnetic moment **m**. $\rho_s$ and $\xi$ are determined from London's equation, $\mathbf{j} = -\rho_s \mathbf{A}$, and its range of validity. This method is particularly accurate at temperatures close to the critical temperature $T_c$. We find weaker $\rho_s$ and longer $\xi$ than existing literature reports, and critical exponents which agree better with expectations based on the Ginzburg–Landau theory.

**Keywords:** superconductivity; iron-based superconductors; magnetism; stiffness; coherence length; Fe$_{1+y}$Se$_x$Te$_{1-x}$

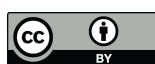

## 1. Introduction

The highest $T_c$ measured in bulk iron-based superconductors (IBSs), in ambient pressure, is 56 K [1], higher than some cuprates, e.g., optinally doped La$_{2-x}$Sr$_x$CuO$_4$. Consequently, they have been at the forefront of research in the solid-state community. Out of all IBSs, the crystalline structure of the FeSe is the simplest. By partially replacing Se with Te atoms, the critical temperature increases up to 15 K, obtained at $x = 0.45$ $y \simeq 0$ in the formula Fe$_{1+y}$Se$_x$Te$_{1-x}$. As summarized by Kreisel et al. [2], the material also possesses surprising properties, such as highly anisotropic electronic properties (nematic effects) and evidence for topologically non-trivial bands and superconductivity. In light of these properties, it is important to characterize Fe$_{1+y}$Se$_x$Te$_{1-x}$ as accurately as possible. Here, we focus on the $x \sim 0.5$ $y \sim 0$ variant (FST), which is available as bulk crystal.

Bulk DC superconducting properties, such as the stiffness $\rho_s$, were measured in this crystal by transverse field muon spin rotation ($\mu$SR) [3,4]. AC measurements were performed by RF tunnel diode [5,6] and cavity perturbation [7,8] techniques. The coherence length of FST with $x = 0.45$ was determined by vortex size $\xi$, using a scanning tunneling microscope (STM) [9] and resistivity measurements [10]. The Cooper pair size $\xi_0$ was evaluated with angle-resolved photoemission spectroscopy (ARPES) [11]. However, due to the presence of Fe in the structure and residual magnetism, the field dependent measurements might not provide a clear insight into the superconducting properties, since the applied field interacts with a magnetic moment in addition to the superconducting currents. In this work, we measure DC superconducting properties in a zero-applied field to avoid contamination from magnetism.

The superconducting stiffness $\rho_s$ is defined via the gauge-invariant relation between the current density **j**, the total vector potential **A**$_{\text{tot}}$ from all sources, and the complex order parameter $\Psi(\mathbf{r}) = \psi(\mathbf{r})e^{i\phi(\mathbf{r})}$ with $\psi(\mathbf{r}) > 0$, according to:

$$\mathbf{j} = -\rho_s \left( \mathbf{A}_{\text{tot}} - \frac{\Phi_0}{2\pi} \boldsymbol{\nabla}\phi \right), \tag{1}$$

where $\Phi_0 = 2\pi\hbar/e^*$ is the superconducting flux quanta,

$$\rho_s = \frac{\psi^2 e^{*2}}{m^*}, \tag{2}$$

is the stiffness, and $e^*$ and $m^*$ are the carrier's charge and mass, respectively. For anisotropic stiffness, see Ref. [12]. When cooling the superconductor (SC) with $\mathbf{A}_{\text{tot}} = 0$ in the London gauge, minimum kinetic energy requires $\boldsymbol{\nabla}\phi = 0$. According to the second Josephson relation, $\phi$ can only change by dissipating energy. Thus, Equation (1) becomes the London equation:

$$\mathbf{j} = -\rho_s \mathbf{A}_{\text{tot}}. \tag{3}$$

This relation holds as long as $\boldsymbol{\nabla}\phi$ does not change. The stiffness, in turn, is related to the penetration depth via:

$$\rho_s = \frac{1}{\mu_0 \lambda^2}. \tag{4}$$

However, every superconductor has a critical current density $j_c$, determined by the penetration depth $\lambda$ and coherence length $\xi$. When $\mathbf{A}_{\text{tot}}$ exceeds a certain value, it is worthwhile for the SC to change $\boldsymbol{\nabla}\phi$ so as to keep $j$ below $j_c$ everywhere in the SC. According to the Josephson equation, dynamic changes in $\phi$ lead to voltage, which, when combined with current, result in power and energy dissipation in the process. When this happens, the relation between $\mathbf{j}$ and $\mathbf{A}_{\text{tot}}$ is no longer linear and the system's rigidity breaks. We used these properties to measure both $\rho_s$ and $\xi$ as a function of temperature in FST.

## 2. Experimental Setup

In the experiment, a ring-shaped SC cut out of a single crystal is used, shown in Figure 1a, with a femtosecond laser. The ring is presented in panel (b). Since FST is brittle, the ring is not perfect. However, as we argue below (in Section 4.1), the smallest outer radius and height count for our analysis. The ring is pierced by a long excitation coil (EC). These parts are shown in panel (c). The excitation coil, ring, and second-order gradiometer are surrounded by a main coil, as in panel (d). The main coil is used to zero the field to less than 0.1 μT on the ring, and for field-dependent measurement. Details of the dimensions of the different parts are given in the figure's caption. EC current $I_{\text{ec}}$ generates a vector potential $\mathbf{A}_{\text{ec}}$ on the ring, nominally without a magnetic field $H$. This vector potential is responsible for persistent rotational current in the superconducting ring. This rotational current produces its own vector potential $\mathbf{A}_{\text{sc}}$ and a magnetic moment. The vector potential in Equation (1) is $\mathbf{A}_{\text{tot}} = \mathbf{A}_{\text{ec}} + \mathbf{A}_{\text{sc}}$. The sample's magnetic moment $m$ is detected by vibrating the ring with the EC rigidly relative to the gradiometer. This mode is called vibrating sample magnetometer (VSM) mode. It utilizes a lock-in amplifier to measure the SQUID output voltage at twice the vibration frequency. This output is proportional to the magnetic flux of a sample through the gradiometer, namely, the vector potential of the sample. It could also be represented by the magnetic moment of the sample.

The gradiometer is composed of two outer loops wound clockwise, and two inner loops wound anticlockwise, see Figure 1d. In that way, we separate the magnetic signal generated by the sample from any other field uniform in space, even if it drifted over time. The gradiometer, main coil, and SQUID are part of the QD-MPMS3 magnetometer.

In principle, $\mathbf{A}_{\text{ec}}$ does not change as the coil vibrates, since there are no EC flux $\Phi_{\text{ec}}$ variations, and the pickup-loop signal is only due to $\mathbf{A}_{\text{sc}}$. In practice, the small signal of the EC is reduced from the measurements as background, (see Section 3.1). The ring's vector potential at a pickup-loop radius $R_{\text{pl}}$ at $z = 0$ is related to $\mathbf{m}$ in the EC direction $\hat{z}$, by:

$$\mathbf{A}_{\text{sc}}(r = R_{\text{pl}}, z = 0) = \frac{\mu_0}{4\pi} \frac{m}{R_{\text{pl}}^2} \hat{\varphi}. \tag{5}$$

where $\hat{\varphi}$ is the azimuth direction.

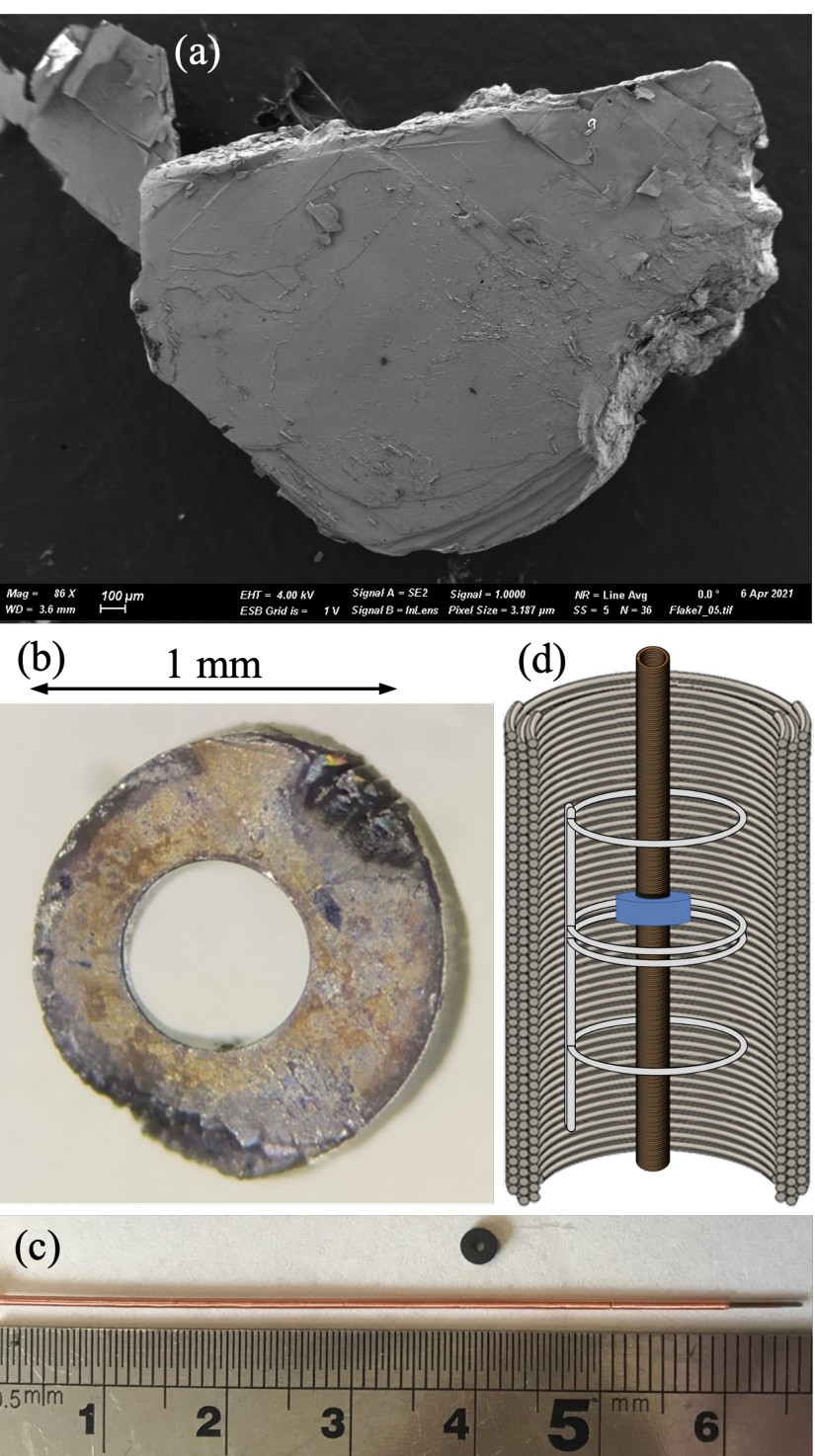

**Figure 1.** Experimental setup: (**a**) A scanning electron microscope image of a single crystal of FST, from which the ring was cut out. (**b**) A microscopic image of the $Fe_{1+y}Se_{0.5}Te_{0.5}$ ring. The sample is not uniform. The minimal height, inner, and minimal outer radii are $h = 0.10$ mm, $r_{in} = 0.26$ mm, and $r_{out} = 0.50$ mm, respectively. (**c**) A copper excitation coil and a superconducting ring beside it. The coil has a length of 60 mm, an outer diameter of 0.25 mm, and 9300 turns in two layers. (**d**) The ring and excitation coil assembly moves rigidly relative to a gradiometer, connected to a SQUID system (not shown), and surrounded by a main coil for field zeroing or field-dependent measurement. The SQUID, gradiometer, and main coil are part of a QD-MPMS3 system.

## 3. Measurements

### 3.1. Stiffness and Critical Current

We cooled the system below $T_c$ with $I_{ec} = 0$. After the temperature has stabilized, we gradually increased $I_{ec}$ while measuring the superconducting ring's magnetic moment. An example of a measurement at $T = 12$ K is presented in Figure 2a-inset. A repetition of this process at different temperatures appears in panel (a). To isolate the superconducting signal, we subtracted the moment of the measurement with zero current, which is due to the ferromagnetic properties of FST and not its stiffness. In addition, we removed the current dependent of the signal above $T_c$. This signal is due to the EC's finite length and asymmetry.

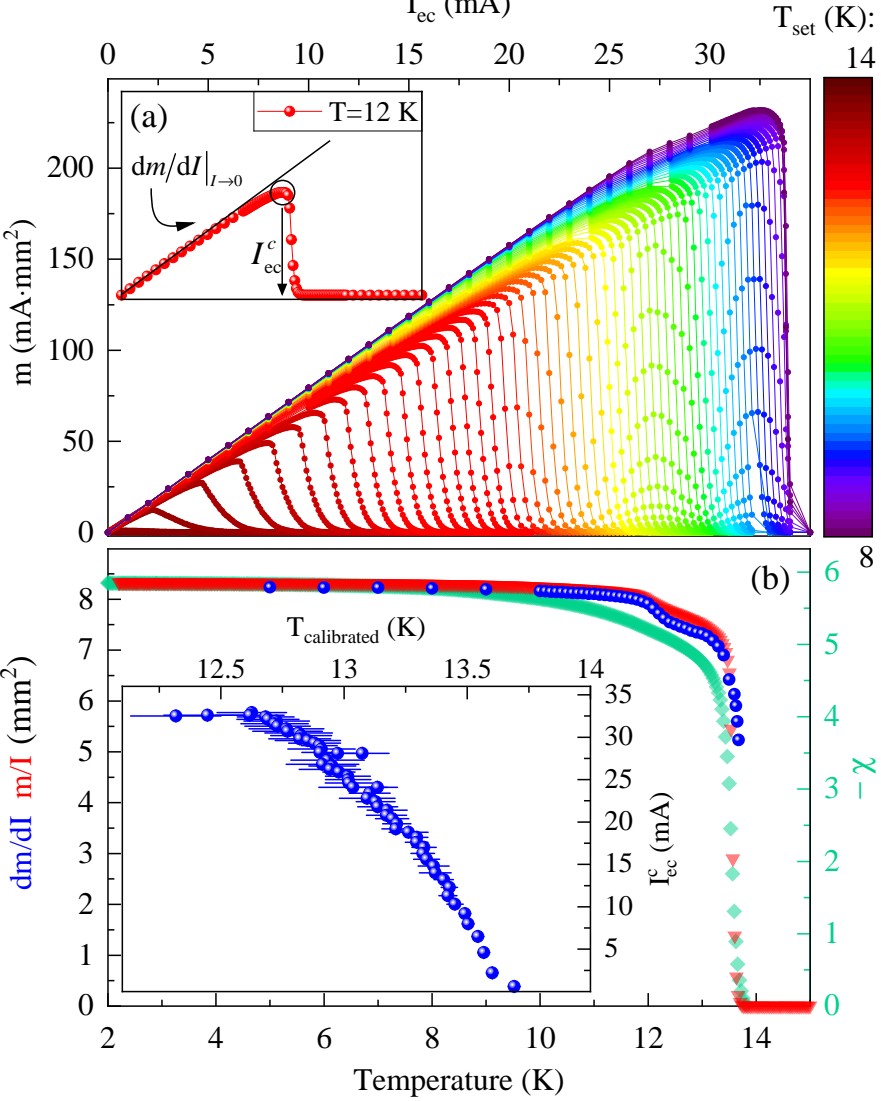

**Figure 2.** Data: (**a**) Stiffness measurements. SC's magnetic moment vs. the current in the excitation coil at different temperatures, indicated by the colors. The inset is focused on the measurement at 12 K. A linear relation is found for low currents. At some critical current value, the signal drops to zero. The blue circles in (**b**) depict the temperature dependence of the linear slope obtained at low currents (far from $I_{ec}^c$) in panel (**a**). (**b**) Critical temperature. SC's current in the EC vs. the temperature (red down-pointing triangles), as described in Section 3.1; measured susceptibility (with a minus sign) vs. the temperature (emerald diamonds) in MKS units in the presence of a magnetic field of 1 mT and without an excitation coil (according to Section 3.2). Inset (**b**) shows the critical currents vs. the calibrated temperature (extracted from the breakpoints in panel (**a**)).

Typical behavior in our measurements, for low currents, is a linear relation between the ring's moment and $I_{ec}$, as expected (see Ref. [13] and Section 4). At some value of $I_{ec}$, which defines the critical current $I_{ec}^c$, this relation breaks. Beyond the breakpoint, the magnetic signal drops sharply instead of the saturation behavior seen previously [14,15]. This drop is a result of two effects: (I) heat produced by the copper EC, which leads to a temperature gradient between the ring and the thermometer and (II) heat produced by energy dissipation as vortices enter the sample and $\phi$ changes dynamically. In fact, when the moment drops to zero, the ring has passed its critical temperature and stops being superconducting. A simple solution to the undesired effect of (I) could have been to use a superconducting coil; however, the $T_c$ of FST is higher than any commercially available superconducting wire. Instead, we calibrated the temperature at the ring position using an open ring. The calibration is discussed in Appendix A.

To extract the stiffness, we fit each $m(I_{ec})$ to a line in a temperature-dependent range due to the variation in the critical current. Such a fit is demonstrated in the inset of Figure 2a. The slope represents $dm/dI$ in the limit $I_{ec} \to 0$. The temperature dependence of the slopes appears as blue circles in Figure 2b. The measurements do not cover all the temperature ranges up to $T_c$, since it becomes exceedingly difficult to define a linear region in the $m(I_{ec})$ data. At a temperature slightly below $T_c$, a knee appears in the temperature dependence of $dm/dI$.

The red down-pointing triangles in Figure 2b measure $m/I$ as a function of $T$. This is done by cooling with $I_{ec} = 10$ mA, turning the current off, and warming while measuring. At $T > 13.45$ K, this current is above $I_{ec}^c$, and such a measurement cannot be used to extract the stiffness near $T_c$. On the other hand, such measurement can be carried out all the way to $T_c$. Interestingly, the knee is observed even with this constant current measurement. It is important to mention that the knee was detected in other FST rings as well. A detailed discussion on the knee is given in Section 7.1.

Finally, in Figure 2b-inset, we present $I_{ec}^c(T)$, corresponding to the moment's maximum, as a function of the calibrated temperature. The large error bars at the low temperatures range are due to the strong current in the coil, leading to a significant temperature gradient and uncertainty in the temperature calibration.

### 3.2. Susceptibility

The emerald diamonds in Figure 2b depict the temperature dependence of the measured, zero-field-cooled (ZFC) susceptibility $\chi = m/(HV_{ring})$, with a field of $\mu_0 H = 0.98$ mT parallel to the axial direction of the ring; $V$ stands for the ring's volume. The specific susceptibility is related to the measured one by:

$$\chi = \frac{\chi_0}{1 + D\chi_0}, \tag{6}$$

where $D$ is the demagnetization factor and $\chi_0$ is the specific susceptibility. For a ring with our geometry, the demagnetization factor equals $D = 0.6$, and if we consider the inner radius of the ring $r_{in} \to 0$, since in ZFC, it is hard for the field to penetrate the ring hole, $D = 0.7$ [16]. With these $D$ values (considering the effective volume of the ring in the latter case), we obtain, at $T \to 0$, $\chi_0 = -1.30$ and $\chi_0 = -1.15$, respectively. $\chi_0 = -1$ is excepted in the case that all of the ring's volume is superconducting. The extra 15% or more in $\chi_0$ could be a result of the irregular shape of the ring. In any case, it indicates that the entire sample is superconducting.

As for the temperature dependence of $\chi$, a sharp transition is observed towards the critical temperature in this measurement $T_c = 13.82$ K, which indicates the quality of the material. Interestingly, in DC magnetization measurements, the knee is not observed.

### 3.3. Hysteresis

To characterize the magnetic properties of the FST sample, we performed a magnetic hysteresis loop measurement, between 2 T and $-2$ T, which is depicted in Figure 3a. This

measurement is performed above the critical temperature, at $T = 15$ K. The opening of a hysteresis loop is an indication of ferromagnetism. Another sign is that the moment of the first point, at $H = 0$, is different from zero. It might be difficult to notice this in the figure. However, this feature makes it possible to detect the sample without applying fields or currents above and below $T_c$, in contrast to non-magnetic materials. Additional properties that can be deduced from this measurement are the magnetization saturation, retentivity (remanence), and coercivity values: $m_{\text{sat}} = 1.58$ A·mm$^2$, $m_{\text{remanence}} = 0.22$ A·mm$^2$, and $\mu_0 H_{\text{coercivity}} = 0.0153$ T, respectively. Although this ferromagnetism is sometimes ascribed to the topological surface state [17], we analyze it as a bulk property. From the magnetization saturation and the magnetic moment of a free Fe ion $m_{\text{Fe}^{2+}} = 5.4\ \mu_B$ or $m_{\text{Fe}^{3+}} = 5.9\ \mu_B$, where $\mu_B$ is the Bohr magneton [18], we can deduce that the fraction of the free iron ions per unit formula in the sample is $y = 0.009$ or $y = 0.008$, respectively. Wang et al. [19], performed inelastic neutron scattering measurements of Fe$_{0.98}$Se$_{0.5}$Te$_{0.5}$ and claimed that $m_{\text{Fe}} = 2.85\ \mu_B$. The corresponding value for the iron fraction in our sample is $y = 0.017$.

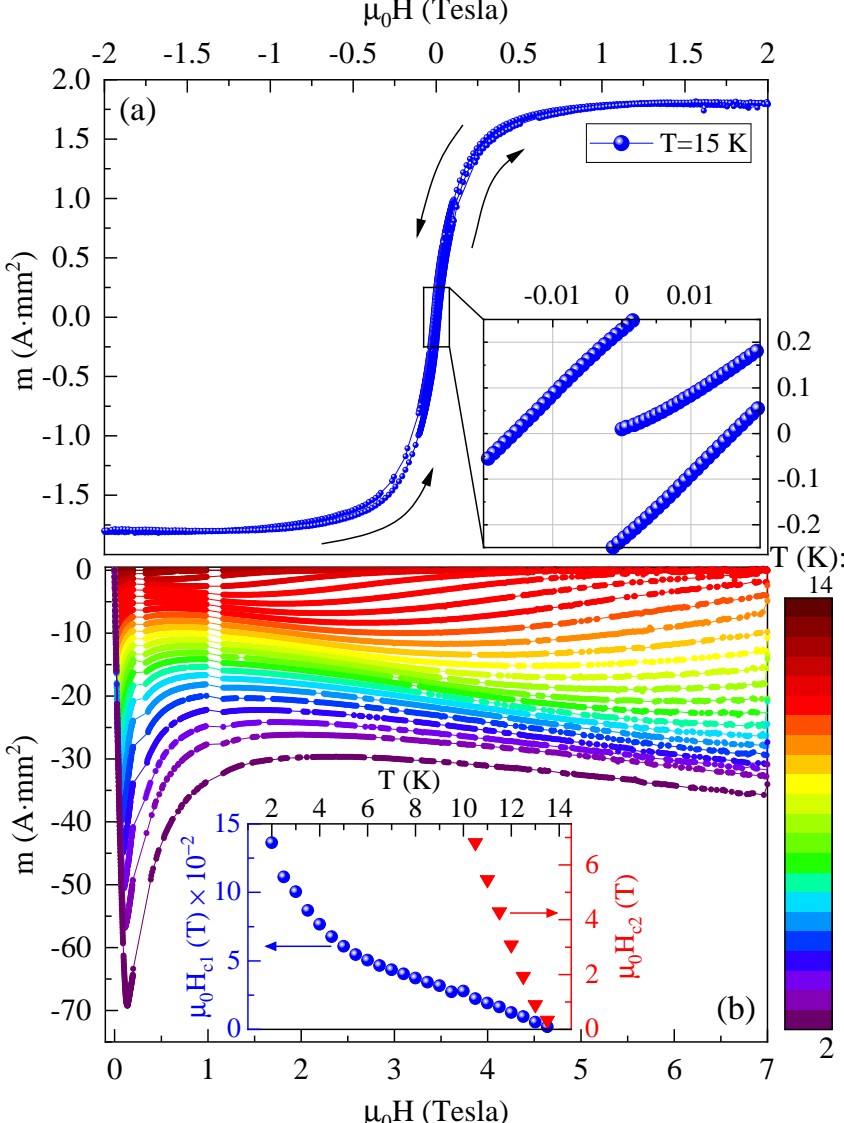

**Figure 3.** Magnetic measurements: (**a**) Magnetic hysteresis loop above the critical temperature. (**b**) $m(H)$ at different temperatures below $T_c$, as indicated by the colors. Inset: The temperature dependence of the critical fields $H_{c1}$ (blue circles) and $H_{c2}$ (red down-pointing triangles) on the left and right Y-axis, respectively.

### 3.4. Critical Magnetic Fields

The response of the superconducting ring to an applied magnetic field at different temperatures below $T_c$ is reflected in Figure 3b. From that measurement, we extract the first and second critical fields, $H_{c1}$ and $H_{c2}$. $H_{c1}$ is defined by the maximum magnitude of the moment for each temperature. A second peak emerges at an intermediate field between $H_{c1}$ and $H_{c2}$, and is attributed to the role of twin boundaries [20]. In principle, $H_{c2}$ is defined by the value of $H$, for which $m = 0$ [21]. However, it is not easy to determine $H_{c2}$ because of the asymptotic behavior of the moment. Therefore, we chose a criterion by which $H_{c2}$ is the field at which the moment is 10% of the second peak magnitude. Below a temperature of 10 K, $H_{c2}$ becomes higher than the maximum field available to us. $H_{c1}$ and $H_{c2}$ as a function of temperature are shown in the inset of panel (b).

## 4. Analysis Model

The analysis of Stiffnessometer data is described in detail in Ref. [13], and is valid for systems with cylindrical symmetry. Here, we provide only the major steps.

### 4.1. Stiffness

In the low-flux regime (low currents in the EC), the magnitude of the order parameter is constant almost all over the superconducting ring and zero outside [13]. Substituting $\mathbf{B} = \boldsymbol{\nabla} \times \mathbf{A}$ and the London equation into ampère's law gives:

$$\boldsymbol{\nabla} \times \boldsymbol{\nabla} \times \mathbf{A}_{\text{sc}} = -\mu_0 \rho_s \mathbf{A}_{\text{tot}}, \tag{7}$$

since on the ring $\boldsymbol{\nabla} \times \mathbf{A}_{\text{ec}} = 0$. In the London gauge, $\boldsymbol{\nabla} \times \boldsymbol{\nabla} \times \mathbf{A} = -\nabla^2 \mathbf{A}$, and the vector potential outside an infinitely long coil is given by:

$$\mathbf{A}_{\text{ec}}(r) = \Phi_{\text{ec}}/(2\pi r)\hat{\varphi}. \tag{8}$$

With Equation (4), we arrive at the partial differential equation (PDE):

$$\nabla^2 \mathbf{A}_{\text{sc}} = \frac{1}{\lambda^2}\left(\mathbf{A}_{\text{sc}} + \frac{\Phi_{\text{ec}}}{2\pi r}\hat{\varphi}\right), \tag{9}$$

where $\lambda = \infty$ outside the SC. Normalizing the spatial variables and vector potential is completed as follows:

$$\mathbf{r}/R_{\text{pl}} \to \mathbf{r}, \quad \mathbf{A}_{\text{sc}}/\mathbf{A}_{\text{ec}}(R_{\text{pl}}) \to \mathbf{A}, \quad \lambda/R_{\text{pl}} \to \lambda, \tag{10}$$

and using cylindrical coordinates where $\mathbf{A} = A(r,z)\hat{\varphi}$, we end up with the following PDE:

$$\frac{\partial^2 A}{\partial z^2} + \frac{\partial^2 A}{\partial r^2} + \frac{1}{r}\frac{\partial A}{\partial r} - \frac{A}{r^2} = \frac{1}{\lambda^2}\left(A + \frac{1}{r}\right). \tag{11}$$

We use the finite element-based FreeFem++ software [22] to solve this PDE for different values of $\lambda$ and the dimension of our FST ring, appearing in the caption of Figure 1. The equation is solved in a box such that $z \in [-L, L]$, and $r \in [0, 8L]$ with $L = R_{\text{pl}} = 8.5$ mm. Dirichlet boundary conditions are imposed.

As shown in Figure 1b, the ring's outer radius is not uniform. However, the solution of the Ginzburg–Landau (GL) equations [13] shows that in the low-flux regime, the current flow is in a layer of width $\lambda$ near the inner rim of the ring, so the system's symmetry is not severely compromised. When the flux through the ring is increased, the current layer retreats toward the outer rim. This retraction ends when the current layer reaches the outer rim. In our case, we assume that it happens at the shortest distance of the outer rim from the center. We use this distance as the outer radius in the PDE (11). Nevertheless, our assumption has not been tested numerically and the impact of a non-perfect ring on the result is not clear yet.

The red line in the inset of Figure 4a depicts the numerical solution of PDE 11. The Y-axis is the normalized vector potential $A$ at the ring's height $z = 0$ and the pickup-loop radial location $r = R_{pl}$. The X-axis is $(R_{pl}/\lambda)^2$ on a logarithmic scale. Normalizing Equation (5) by the vector potential of an infinite coil, we obtain:

$$A_{ec}(R_{pl}) = \frac{\mu_0 n I_{ec}}{2R_{pl}} \sum_i r_{ec,i}^2 ,  \tag{12}$$

where $n$ and $r_{ec,i}$ are windings per unit length in one layer and radius of the $i$th layer, respectively. We obtain the following dimensionless vector potential:

$$A(z = 0, R_{pl}) = \frac{g}{2\pi n R_{pl} \sum_i r_{ec,i}^2} \cdot \frac{m}{I_{ec}} ,  \tag{13}$$

where $m$ is the SC's magnetic moment and $g$ is a geometrical constant on the order of unity.

In reality, the coil is not infinite, and, as a result of cutting and drilling, the ring is not perfect, see Figure 1b. Therefore, the calibration constant $g$ is determined experimentally in two different methods: (1) We compare the saturated value of $A$ from the solution of PDE (11) (see the red line in Figure 4a-inset) to the saturated value of $dm/dI$ [the lowest available temperature of the blue circles in Figure 2b]. This method cannot be used to determine $\lambda(T \to 0)$, since exactly this limit is used for the calibration. Nevertheless, it gives one value for $g$. (2) From the literature, we use low-temperature stiffness value of a similar material to predict $A$ with the PDE solution and compare it to our measured $dm/dI$ at the same temperature to extract a second value for $g$. For this work, the stiffness was taken from Ref. [3]. We found $g_1 = 0.5363$ and $g_2 = 0.5336$ using methods (1) and (2), respectively. We also applied the same calibration methods for a ring-shaped Niobium with similar dimensions and found $g_1 = g_2 = 0.68674(2)$ while using $\lambda(0) = 39$ nm as the literature value for Niobium [23]. Although the two calibration methods give different values for the penetration depth at low temperatures, towards $T_c$, the values converge and almost coalesce, as we demonstrate shortly. In other words, the stiffness determined by the Stiffnessometer is not sensitive to the calibration method once $dm/dI$ is out of the saturation region.

*4.2. Coherence Length*

In the low-flux regime $\Phi_{ec}/\Phi_0 \ll r_{in}^2/\lambda\varepsilon$, and for $\lambda \ll r_{out} - r_{in}$ and $h$, where $h$ is the ring's height, deep inside the ring $A_{tot} = 0$; hence, $A_{sc} = -A_{ec}$. In other words, the applied flux is matched by the flux generated by the ring in the hole. For $\Phi_{ec}/\Phi_0 > r_{in}^2/\sqrt{8}\xi\lambda$, the current necessary to produce $A_{sc}$ at $r_{in}$ exceeds the local critical current [13]. Then, it is energetically preferable for the order parameter magnitude to gradually diminish in the inner rim of the ring. Consequently, the superconducting ring hole effectively grows, and an effective inner radius $r_{eff}$ is established. At even higher flux, $r_{eff}$ approaches $r_{out}$, and the SC is no longer able to expel the applied flux, namely, to cancel $A_{ec}$. This happens at a critical flux [13]:

$$\frac{\Phi_c}{\Phi_0} = \frac{r_{out}^2}{\sqrt{8}\xi\lambda} .  \tag{14}$$

While the derivation of Equation (14) is in the limit $\xi \ll \lambda \ll r_{out} - r_{in} \ll h$, we believe it is valid for $\lambda \ll r_{out} - r_{in}$ and $\lambda \ll h$ separately.

For $\Phi > \Phi_c$, vortices are expected to penetrate from the inner rim towards the outer one so that the SC's moment no longer grows with amplification of $I_{ec}$. These vortices are manifested in $\nabla\phi$ variations.

### 5. Data Analysis

Equation (13) relates the linear slope of the $m(I)$ measurements, shown by blue circles in Figure 2b, to the numerical solution of the PDE. The blue open circles in Figure 4a-inset represent the converted points using $g_2$. Each of those points belongs to a different temperature and gives a unique stiffness value. The temperature dependence of $\lambda$ is presented on a linear scale in Figure 4a, right Y-axis, and of $\lambda^{-2}$ on a logarithmic scale in Figure 4b, for the two different $g$ values. The difference between the two calibration methods is revealed in both subfigures, but they are minute at $T \to T_c$. The linear regression towards the critical temperature on the logarithmic scale represents the critical exponent $n_\rho$, according to the power law:

$$\rho \propto (1 - T/T_c)^{n_\rho} , \tag{15}$$

where $n_\rho = 0.91 \pm 0.02$. This relation describes the data well from the knee temperature 12.4 K all the way to $T_c$. For comparison, the μSR measurements of $1/\lambda^2$ [3,4] are also added to Figure 4b, and their $n_\rho = 0.53 \pm 0.04$. It should be pointed out that all techniques agree that $\lambda(T = 0) \sim 0.5$ μm, but from the tunnel diode technique, $\lambda(T = 0.9T_c) \sim 2$ μm [6], which is longer than μSR, but shorter than the Stiffnessometer.

Based on the stiffness and the critical current in the inset of Figure 2b, we extract the coherence length using Equation (14) and the calculated flux in the coil. The results are depicted on a linear scale in Figure 4a and on a logarithmic scale in Figure 4c. Again, we fit the data to the power law:

$$\xi^{-1} \propto (1 - T/T_c)^{n_\xi} . \tag{16}$$

We found $n_\xi = 0.41 \pm 0.02$. The deviation from the linear regression at high temperatures may be a result of analysis failure, since the penetration depth is no longer much smaller than the ring's height ($\lambda \ll h$). At low temperatures, we associate the deviation with heating caused by the strong current in the excitation coil, which cannot be accurately accounted for by the temperature calibration.

The alternative determination of $\xi$ is from $H_{c2}$ [21] according to the equation:

$$\mu_0 H_{c2} = \frac{\Phi_0}{2\pi \xi^2(T)} . \tag{17}$$

$\xi$, determined from $H_{c2}$, is presented on a linear scale with black squares in Figure 4a and $1/\xi$ is presented on a logarithmic scale in panel (c) of the same figure for comparison. Here, we also fit the data according to Equation (16) and obtained $n_\xi = 0.60 \pm 0.03$.

For further comparison, low-temperature measurements of $1/\xi$ from other methods have been added to Figure 4c (star-shaped): resistivity [10], ARPES [11], and STM [9]. The resistivity measurement is, in fact, an $H_{c2}$ measurement, and the result obtained is close to the one obtained by the magnetization method ($\xi_{H_{c2}}/\xi_{\text{Res}} = 1.6$ at $T = 0$). The ARPES value $\xi_0$ is related to the GL $\xi$ at $T = 0$, $\xi(0)$, by a factor of 0.74 [21] (Equation 4.24). The same factor was taken into account when converting the STM result. Unlike the $H_{c2}$ measurements, the results from the other two methods are closer to the linear regression of the Stiffnessometer method ($\xi_{\text{Stiff}}/\xi_{\text{ARPES}} = 1.7$ and $\xi_{\text{Stiff}}/\xi_{\text{STM}} = 1.9$ at $T \to 0$).

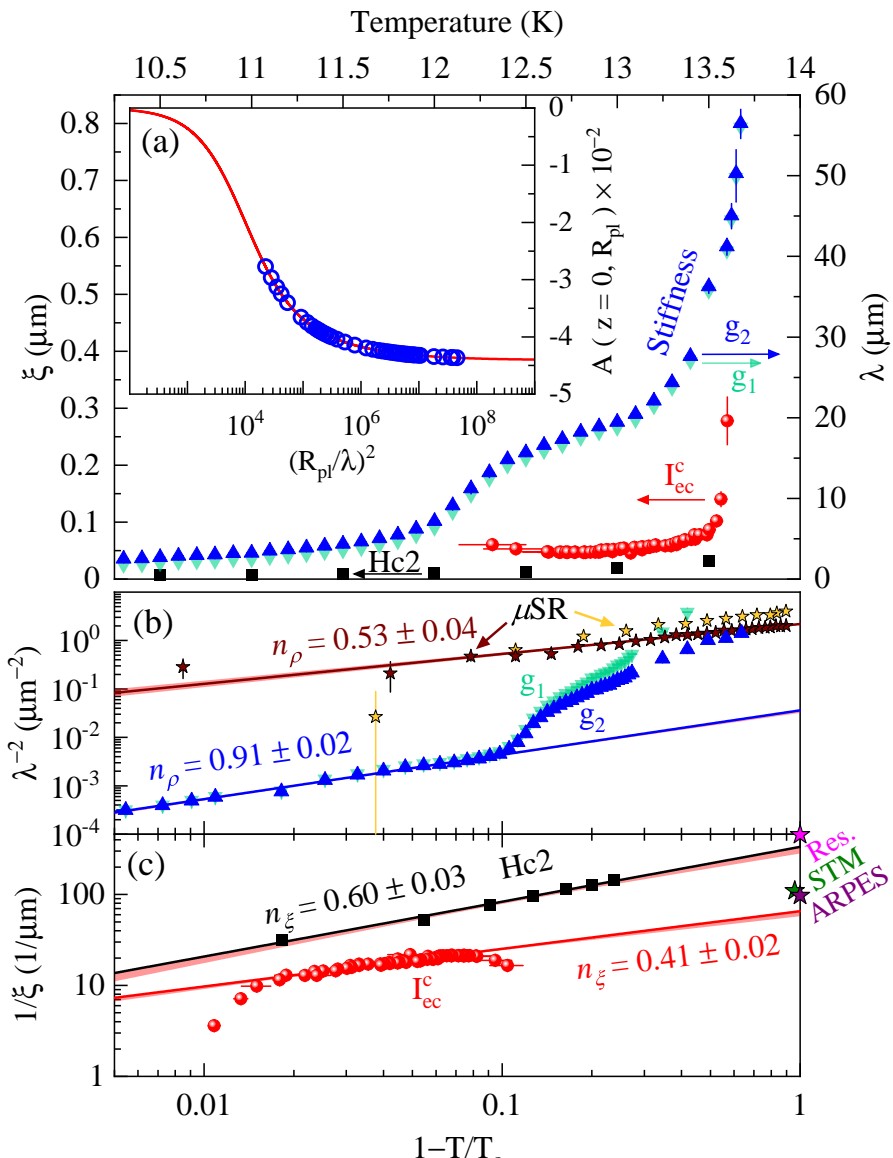

**Figure 4.** Penetration depth and coherence length. (**a**) The right Y-axis shows the penetration depth as a function of the temperature in blue and emerald triangles, for the different calibration methods. The left Y-axis shows the temperature dependence of the coherence length. The red-circles are taken from the critical current measurement in Figure 2b-inset through Equation (14) with the measured $\lambda$, and the black-squares are from the second critical field in Figure 3b-inset with Equation (17). Panels (**b**,**c**) are log-log plots of the stiffness $\lambda^{-2}$ and $1/\xi$ vs. $1 - T/T_c$, respectively. The linear regression represents the critical exponents according to Equations (15) and (16), respectively. Earlier stiffness measurements using the μSR method have been added to (**b**) in brown [3] and yellow [4] stars. The same power law is fitted to this data. For comparison, we add to (**c**) asterisks reflecting the measurements of $1/\xi$ from the resistivity method [10] in magenta, ARPES [11] in purple, and STM [9] in green.

## 6. Reproducibility and Origin of the Knee

To examine reproducibility, we investigated more than one ring cut from different crystals of FST. A comparison between different rings from other crystals appears in Figure 5. The figure shows the normalized, and shifted (for clarity), SC moments as a function of the temperature in two cases: panel (a) with a current in the EC and zero-applied field and panel (b) with an applied field and no EC current. The applied currents and fields are in the range [5, 10] mA and [0.1, 3] mT, respectively, but not necessarily equal for different rings.

The main ring of this research is 1. The knee temperature and sharpness vary from ring to ring (Figure 5a). Interestingly, multiple knees appear in the standard, in-field measurement of ring 2 in panel (b).

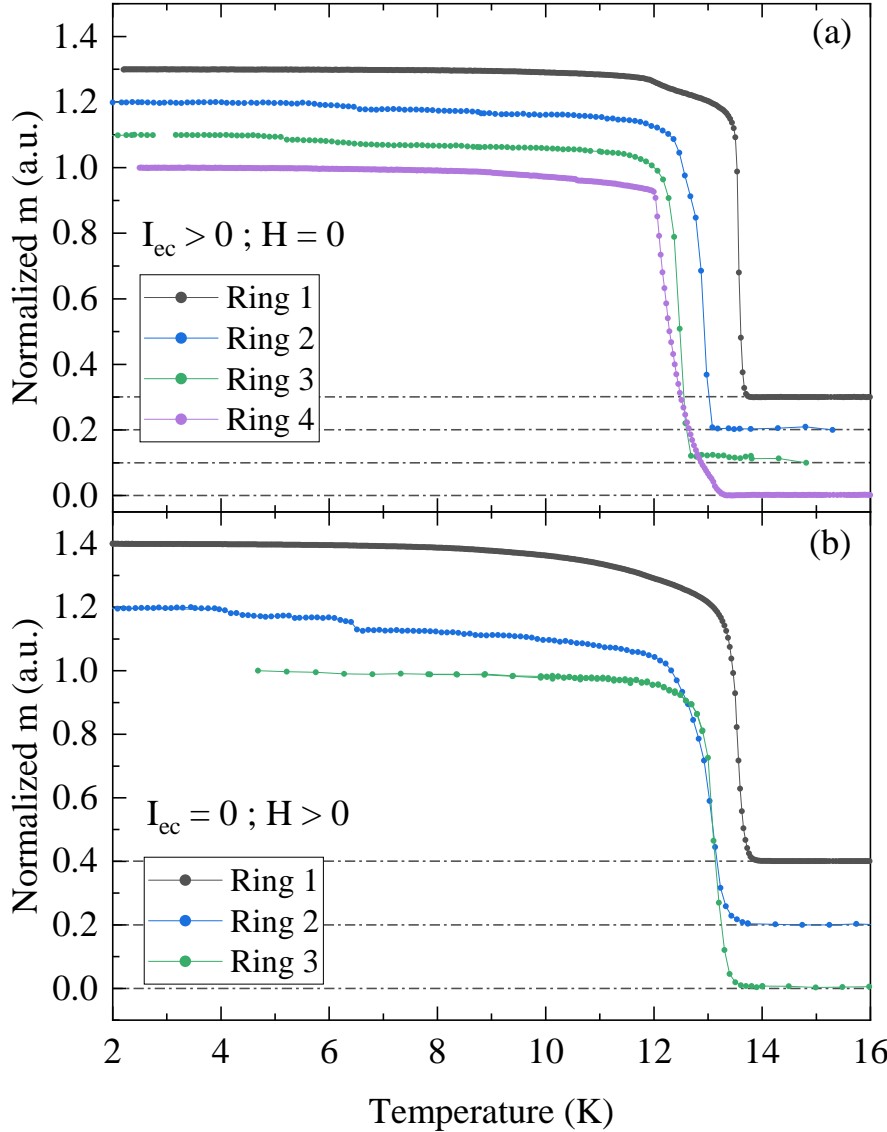

**Figure 5.** Reproducibility. Normalized magnetic moment $m/m(T \to 0)$ vs. temperature for different rings. (**a**) in the presence of current in the excitation coil, as described in Section 3.1. (**b**) in the presence of an applied field perpendicular to the ring. The central ring of this research is 1. An offset is added for clarity.

It could be that the knee originates from the interaction of the superconducting order parameter with the underline ferromagnet. To test this possibility, we measure the SC's magnetic moment vs. temperature in the presence of an applied field (ZFC) in the direction of the EC, with and without current in the coil. The raw data are shown in the inset of Figure 6. The difference between the two measurements is presented in Figure 6. For comparison, the measurement with the current only is also displayed. The knee appears at the same temperature with and without the field.

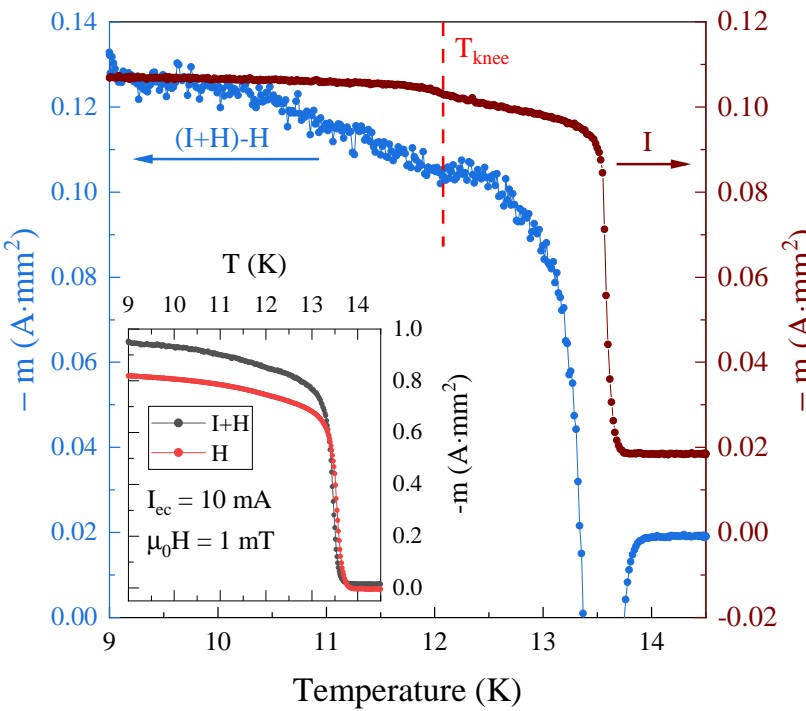

**Figure 6.** The knee's field dependence. Temperature dependence of the difference between the ring's moment measured with the excitation coil current and field, to a measurement with the field only (blue circles, left Y-axis). The brown circles on the right Y-axis are the measurement in the zero-field current in the excitation coil. The data are shifted for clarity. The knee is at the same temperature regardless of the field. The inset shows the ring's moment vs. the temperature in the presence and absence of an applied field (1 mT) and current in the EC (10 mA). The data in the main panel are obtained by subtracting the two datasets in the inset.

## 7. Discussion

We discuss the two major observations of this work, the knee and the critical exponents, and examine the relation between the measured quantities.

### 7.1. The Knee

Mukasa et al. [24] presented the nematic transition temperatures $T_s$ as a function of the Tellurium composition in $Fe_{1+y}Se_xTe_{1-x}$. The lowest temperature measured by X-ray diffraction is 13.3 K, and there are no measurements close to the mid point $x = 0.5$. Nevertheless, extrapolation of their data suggests that $T_s$ and $T_c$ cross each other near $x = 0.5$ and that $T_s$ might drop below $T_c$. Perhaps, nematic order is the origin of the knee. Alternatively, Peng Zhang et al. [25], suggested the existence of surface superconductivity in FST. We speculate that this might lead to two different SC $T_c$s, one for the bulk and one for the surface. FST is also known to have multiple Fermi surfaces. It could be that the knee is a result of the different temperature dependences of the SC's order parameters on different bands.

Finally, there is always the possibility that the knee is a result of the geometrical imperfection of the ring. Such imperfections are difficult to account for in numerical simulations.

### 7.2. Critical Exponents

The GL theory assumes, and BCS theory predicts, a linear temperature dependence of $\psi^2$. According to Equation (2), this leads to the prediction that $n_\rho = 1$. Our finding is not exactly as expected, but it is closer to unity than the results of μSR added to Figure 4b. It should be pointed out that the μSR measurements are performed in a fixed magnetic

field, which becomes higher than $H_{c2}$ as one approaches $T_c$. The discrepancy between techniques could also result from an interaction between the applied field and the underlined ferromagnet, as mentioned before. Similarly, standard GL predicts $n_\xi = 0.5$. In this case, $\xi$, determined by the Stiffnessometer, and $H_{c2}$ are equally far from the expected value.

If we relax the linear assumption, the GL theory also predicts $n_\rho/n_\xi = 2$. We find:

$$n_\rho/n_\xi = 2.22 \pm 0.12\,. \tag{18}$$

The result obtained from the μSR and $H_{c2}$ methods gives $n_\rho^{\mu SR}/n_\xi^{H_{c2}} = 0.88 \pm 0.08$, far from the GL expected value.

### 7.3. First Critical Field

The first critical field, $H_{c1}$, is related to $\lambda$ and $\xi$ [26] via:

$$\mu_0 H_{c1} = \frac{\Phi_0}{4\pi\lambda^2} \ln \frac{\lambda}{\xi}\,. \tag{19}$$

An attempt to test this equation fails severely regardless of the experimental method used to determine the different quantities. Bendele et al. [4] addressed this problem by considering the demagnetization factor $D$. They introduced the following equation:

$$B = \mu_0(m/V + H_{\text{int}})\,, \tag{20}$$

where $H_{\text{int}} = H_{\text{ext}} - D \cdot m/V$, $H_{\text{int}}$, and $H_{\text{ext}}$ are the internal and externally applied field, respectively, $\mu_0 H_{c1} \to B$ in Equation (19). This calculation is very sensitive to the ring's volume and $D$ accuracy. In Section 3.2, we considered two options for $D$. If we adopt the disk option, we obtain a much smaller $B$ than measured. If we consider the ring option, we find a negative $B$ value. Sometimes, an additional constant is considered in Equation (19), which includes the effect of the hard core of the vortex line [4,27,28], but in our case, this effect is negligible. Once again, we speculate that the failure of Equation (19) is a result of the underlining ferromagnetism in FST.

### 8. Conclusions

We developed a method, ideal for magnetic superconductors close to $T_c$, to measure both the penetration depth $\lambda$ and coherence length $\xi$. For FST, we find that $\lambda$ and $\xi$ are longer than previously reported and their temperature dependence agrees better with the GL predictions. A second transition, which looks like a knee, is observed at a temperature below $T_c$ in the stiffness measurements. Further experiments are required to determine whether this transition is due to either nematic order, surface superconductivity, multiple Fermi surfaces, or a simple geometrical effect.

**Author Contributions:** A.P. and I.M. assembled the experiment. A.P. performed the measurement and data analysis. A.P. and A.K. wrote the paper. All authors have read and agreed to the published version of the manuscript.

**Funding:** This research was supported by the Israel Science Foundation, personal grant number 3875/21, and the Nevet grant, Russel Berrie Nanotechnology Institute, Technion.

**Data Availability Statement:** All the data is presented in the paper and Appendix A.

**Acknowledgments:** We thank Amit Kanigel and Avior Almoalem for the sample. We are grateful to the nano-center at Tel-Aviv University for the use of their femtosecond laser cutter.

**Conflicts of Interest:** There is no conflict of interest.

### Appendix A. Temperature Calibration

Due to the heat produced by the current in the EC, a temperature gradient is developed between the ring and the thermometer, that is, the actual temperature of the sample $T$ and

the temperature recorded by the chamber thermometer $T_{\text{ch}}$ are not the same. Our goal is to determine the sample temperature $T$ corresponding to each critical current $I_{\text{ec}}^{c}$ based on the chamber temperature $T_{\text{ch}}$.

The calibration process is completed by measuring the temperature dependence of the magnetic moment in the presence of an applied field of $\mu_0 H \approx 1$ mT (ZFC), similarly to Section 3.2. However, this time, we use a disconnected FST ring and repeat the measurement for different $I_{\text{ec}}$ values. The critical current values from Figure 2b-inset have been chosen to improve the accuracy.

The current in the EC heats the sample, but cannot generate a persistent current in the ring due to the disconnection. Nevertheless, there are two additional contributions of the EC current to the signal, and both are consequences of its finite length. A good way to understand them is from the EC signal in Figure 2-inset and Figure 3 in Ref. [14]. (I) The second-order gradiometer is insensitive to any field uniform in space, but even around its center, the EC signal is not totally uniform, mostly due to asymmetry of the coil (e.g., wires enter the coil from one side only). This contribution is identified from the measurement above $T_c$ and subtracted. The measurement results after this subtraction appear in Figure A1a. (II) A field leakage from the EC, altering the field in the sample and the sample's moment accordingly. This field leakage could be partially canceled by measuring the moment in two current directions, as presented in Figure A2a. The difference between measurements increases with the current, while the zero-current measurement stays in the middle. Averaging over both directions reduces the deviations due to field leakage, as in Figure A2b. Notably, the magnitude of the field leaking from the coil at a current of 10 mA is estimated to be 0.03 mT.

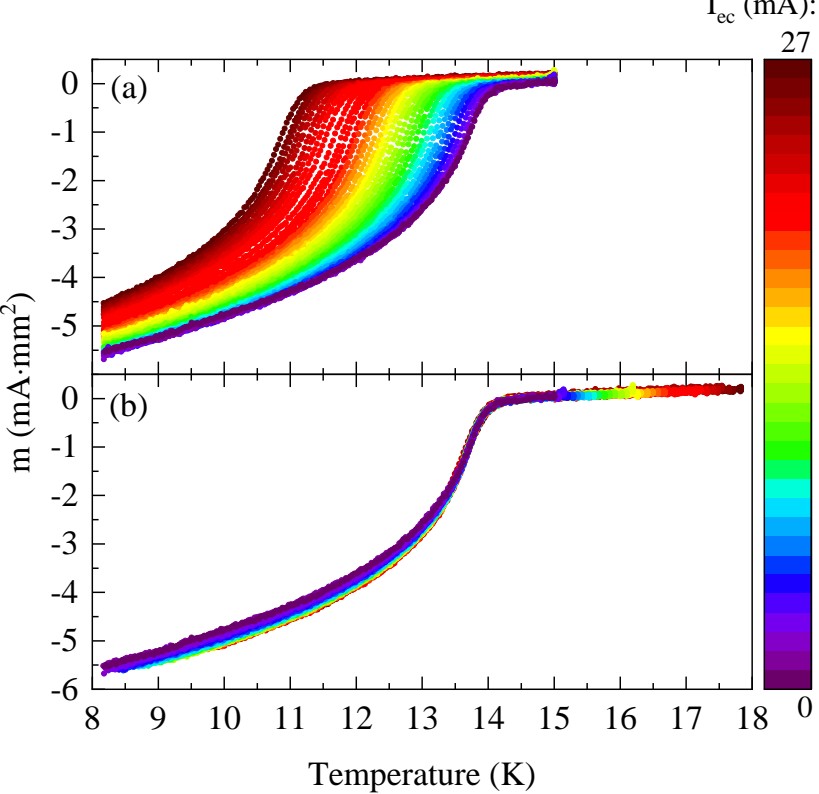

**Figure A1.** Temperature Calibration. Temperature dependence of the magnetic moment of a disconnected FST ring in the presence of a magnetic field, repeated for different $I_{\text{ec}}$, as indicated by the colors. (**a**) Before calibration. (**b**) After calibration.

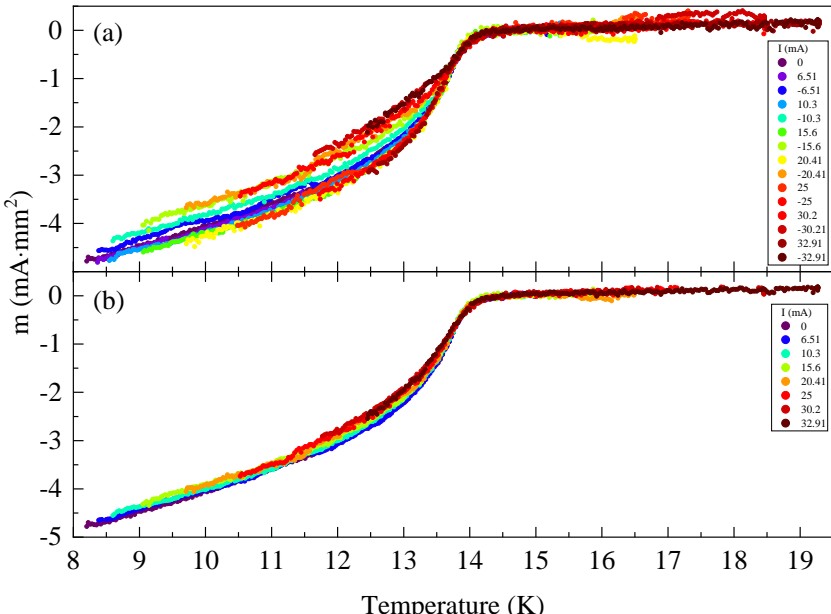

**Figure A2.** The influence of the leaking field from the excitation coil on the measurements.
(**a**) Calibrated measurements in the presence of positive and negative current values, as indicated by
the colors. (**b**) Averaging over the directions of the currents in (**a**).

Once these contributions are eliminated, we search for the temperature correction,
$\Delta T$, for which $m(I_{ec}, T_{ch} + \Delta T)$ collapses onto the one without the current $m(0, T)$ at the
steepest part of the measurement's slope, as seen in Figure A1b. The collapse is best when
close to $T_c$, but the correction is suitable for a wide range of temperatures. The relation
generated between $I_{ec}$ and $\Delta T$ is given in Figure A3-inset.

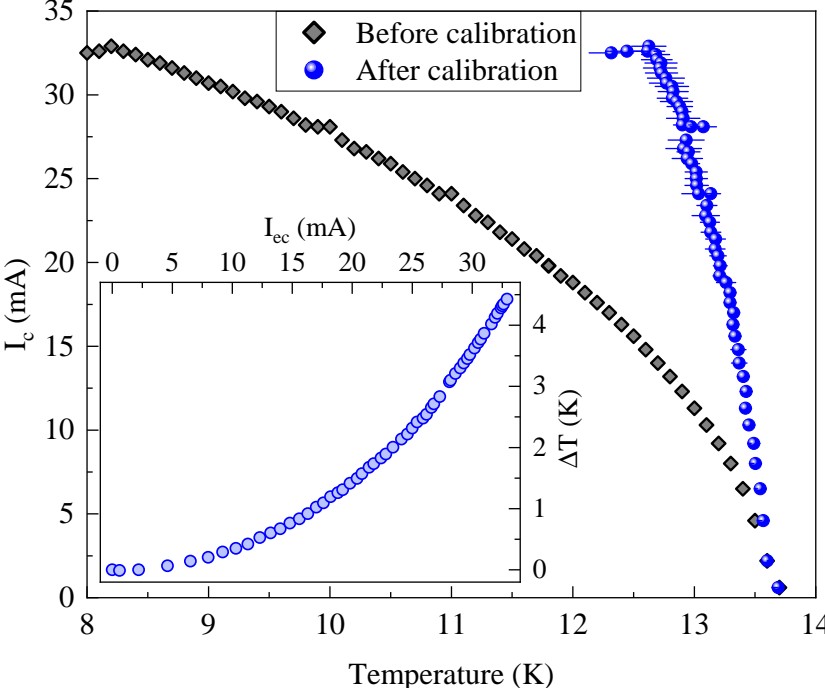

**Figure A3.** Critical currents before and after the calibration. Critical current vs. the temperature
before the calibration in gray diamonds and after in blue circles. The inset shows the temperature
correction $\Delta T$ vs. the current in the excitation coil. The relation is approximately parabolic.

After the temperature correction is set for each current, we compare the measurement with the current to the one without. The error in the temperature correction is estimated by the temperature difference between points with the same moment value from both measurements. An example appears in Figure A4. The errors depend on the current in the coil and temperature. Finally, in blue circles in Figure A3, we present the SC critical current $I^c_{ec}$ as a function of the calibrated temperature $T$ with error bars.

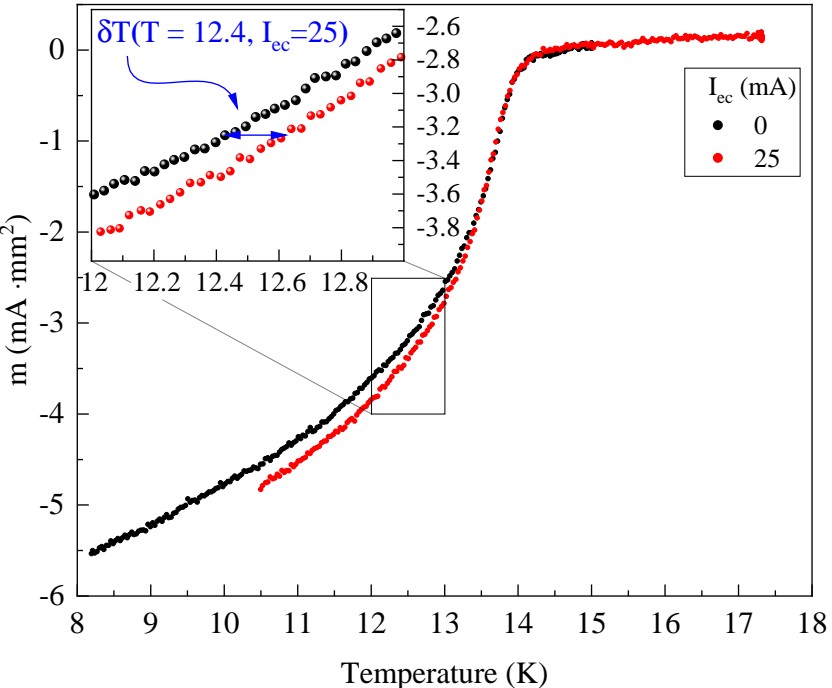

**Figure A4.** Estimation of the temperature calibration process errors. Measurements after temperature calibration (from Figure A1b) without current in the excitation coil in black circles and with a current of $I_{ec} = 25$ mA in red circles. The error is estimated by the temperature difference between two points with the same moment value. It is represented by $\delta T$ and depends on the current and the temperature.

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
