# Peer review of "Superconducting Stiffness and Coherence Length of FeSe0.5Te0.5 Measured in a Zero-Applied Field"

_condensedmatter, doi:10.3390/condmat8020039_

Round 1

Reviewer 1 Report

In the work under review, the authors revisit the measurement of the superconducting stiffness and coherence length in the material FeSe0.5Te0.5 with a different experimental technique than used in previous investigations. The method is based on having a ring of the superconducting material of known geometry, but does not rely on the application of an external field which could make exact measurements difficult in materials that additionally exhibit magnetism. In result, the authors find deviations from earlier measurements, i.e. smaller superfluid density and therefore longer coherence length and analyse their results in view critical exponents which seem to follow better the mean field behavior compared to the data in the literature.

In summary, the work establishes a different dataset for the measured quantities; open questions on discussing the differences or evaluation of the reliability of the experimental method might be subject to further research. The work therefore merits publication, some open questions should however be addressed to improve the manuscript and educate the reader about possible shortcomings of the investigations.

Comments on the manuscript in the order these appear:

1) First sentence: Connecting the work to the area of high temperature superconductivity is certainly relevant, but it is maybe a bit misleading to compare the highest Tc of IBSs to some lower Tc in not “optimal” cuprates. Please try to reformulate for better scientific accuracy.
2) In the introduction the authors refer to “residual magnetism” in FeSe0.5Te0.5. Can you provide a reference for magnetism. Is it bulk or surface? See also recent work: Phys. Rev. Lett. 130, 046702 – (2023)
3) Eq. (2) seems only to hold for the free electron gas with spherical Fermi surface. Please comment on how this can be modified for anisotropic superconductors, for example by including a mass tensor in an appropriate way.
4) paragraph after Eq. (4): What is meant with “dissipating energy” in this context? Is “dissipation” in view of a non-reversible process meant or rather the increase of the kinetic energy upon allowing for a phase winding, i.e. supercurrents?
5) page 2, section IIIA: It is written “while measuring the SC’s magnetic moment”. Can you clarify which quantity is actually measured to make the reader aware which assumptions are entering this approach?
6) The authors mention that their experiment sees something else than in in Refs. 13,14. Any explanation why?
7) Page 4, first paragraph: In the fitting procedure, it is assumed that the excess Fe atoms behave as free Fe ions. This is probably too simplified; also using the moment from the neutron experiments might be error prone. Have the authors done any other measurement for deducing the excess iron concentration?
8) Page 4, section “D”: The authors define “Hc2 as the value of H where m=0”. Isn’t this a bit misleading since one main obstacle is that the system can exhibit magnetization from local moments unrelated to supercurrents, so this seems not a very accurate or direct deduction.
9) page 5, paragraph after Eq. (13): Does also the method “(1)” use a lambda(T-> 0) from the literature? So is it true that only “relative” measurements can be done (which is still a valuable approach, but might have systematic errors if the used literature value of lambda is inaccurate. Please comment.
10) Eq. (14): Can you comment on which assumptions on thee vortex state or pinning effects have been used to derive this. Is it correct for an isotropic superconductor only? How is it modified for anisotropic ones, or dependence on the vortex lattice structure (square vs. triangular).
11) page 6, section VI: In Fig. 5, it is referred to the “normalized SC moment”. Can you specify the normalization (and indicate the shift by showing the “0” for each of the data sets)?
12) page 7, paragraph in “The Knee”: Please clarify what is meant with “This might lead to two different Tc’s...” Can you indicate which theoretical work proposes different critical temperatures? The emergence of surface states for topological superconductors is by construction bound to the critical temperature of the bulk topological superconductor since the presence of the bulk topological state forces the surface state to exist. Please clarify or explain.
13) page 7, same paragraph. What is meant with “order parameters on different surfaces”? You refer to “Fermi surfaces” here? Then maybe “order parameters on different bands” might be more appropriate.
14) Can you define the critical exponents with the appropriate behavior of rho and xi close to Tc? Instead of referring to “GL theory assumes a linear temperature of ...”, one could more generally state that the linear behavior of |\psi|^2 (if the latter is associated with the order parameter squared) is a result of the BCS (mean field) calculation where the critical exponent of the order parameter is 0.5
15) page 8, conclusion section: If understood correctly, one input for the data analysis is the simulation of the shape anisotropy for a given ring geometry to connect the measured signal to the magnetization of the material. However, this assumes that all volume of the ring is superconducting at T->0 and close to Tc where the important measurements are done. The latter might however be altered if the material is slightly inhomogeneous for example by slight changes of the impurity concentration which could easily generate a slightly smaller Tc in some regions such that the ring geometry of the superconducting material would be different (i.e. part of the ring is non-superconducting). Can the authors comment on this possibility and estimate/calculate in which direction the measured quantities would deviate from such an effect?

Author Response

We thank review 1 for the very detailed report. We address each comment individually.

1) First sentence: Connecting the work to the area of high temperature superconductivity is certainly relevant, but it is maybe a bit misleading to compare the highest Tc of IBSs to some lower Tc in not “optimal” cuprates. Please try to reformulate for better scientific accuracy.

Reply> The Tc of optimal LSCO is only 39K. IBS have higher Tc. We added the LSCO example.

2) In the introduction the authors refer to “residual magnetism” in FeSe0.5Te0.5. Can you provide a reference for magnetism. Is it bulk or surface? See also recent work: Phys. Rev. Lett. 130, 046702 – (2023)

Reply> We analyze the magnetism we detected as being bulk. In the new version we added a reference to the above paper indicating that there are other interpretations of this magnetism.

3) Eq. (2) seems only to hold for the free electron gas with spherical Fermi surface. Please comment on how this can be modified for anisotropic superconductors, for example by including a mass tensor in an appropriate way.

Reply> Our group has a paper on anisotropic stiffness (I. Kapon, et al, Nature Communications 10, 2463 (2019).). In the new version we cite this paper after equation 2.

4) paragraph after Eq. (4): What is meant with “dissipating energy” in this context? Is “dissipation” in view of a non-reversible process meant or rather the increase of the kinetic energy upon allowing for a phase winding, i.e. supercurrents?

Reply> We extended the explanation to: “According to the Josephson equation, dynamic changes in $\phi$ lead to voltage, which, when combined with current, result in power and energy dissipation in the process.”

5) page 2, section IIIA: It is written “while measuring the SC’s magnetic moment”. Can you clarify which quantity is actually measured to make the reader aware which assumptions are entering this approach?

Reply> The question is not clear to us. The SC ring carries a current. Current times area of the ring is magnetic moment. We measure this moment. In the new version we added the word ring.

6) The authors mention that their experiment sees something else than in in Refs. 13,14. Any explanation why?

Reply> The experimental observation is that once the order parameter is destroyed by the excitation coil current it never recovers. In Ref. 14 it seems that null order parameter simply allows for vortices to enter, and it recovers and superconductivity persists. We explain in the manuscript that this might be an experimental artifact due to heating of the sample once vortices move into it.

7) Page 4, first paragraph: In the fitting procedure, it is assumed that the excess Fe atoms behave as free Fe ions. This is probably too simplified; also using the moment from the neutron experiments might be error prone. Have the authors done any other measurement for deducing the excess iron concentration?

Reply> We have not checked excess iron concentration. We had no reason to think that neutron experiment might be erroneous.

8) Page 4, section “D”: The authors define “Hc2 as the value of H where m=0”. Isn’t this a bit misleading since one main obstacle is that the system can exhibit magnetization from local moments unrelated to supercurrents, so this seems not a very accurate or direct deduction.

Reply> The data of M vs. H in the superconducting state is presented after M vs. H at T>Tc is subtracted. Therefore, the M vs. H presented in our manuscript is purely from the superconductor.

9) page 5, paragraph after Eq. (13): Does also the method “(1)” use a lambda(T-> 0) from the literature? So is it true that only “relative” measurements can be done (which is still a valuable approach, but might have systematic errors if the used literature value of lambda is inaccurate. Please comment.

Reply> Method 1 does not use literature values. It uses the saturation value of the PDE solution and the saturation value of the measurement (at T->0); g is their ratio. This method prevents us from claiming that we know what is the stiffness at T=0 since the data is used for calibration. The second method uses a measurement value at a particular temperature and compares to the literature at the same particular temperature. In this method we can tell what is the stiffness in all other temperatures including the temperature used for calibration (which is the literature value). Since both methods lead to the same stiffness values at T->Tc, we claim that we can accurately measure the stiffness at this temperature range.

We changed the text to “nevertheless, it gives one value of g; (2) We use a literature value of a low temperature stiffness of similar material to predict A with the PDE solution and compare it to our measured dm/dI at the same temperature to extract a second value of g.

10) Eq. (14): Can you comment on which assumptions on thee vortex state or pinning effects have been used to derive this. Is it correct for an isotropic superconductor only? How is it modified for anisotropic ones, or dependence on the vortex lattice structure (square vs. triangular).

Reply> Equation 14 is correct only if the system has cylindrical symmetry. It is not correct for anisotropic system (unless the anisostropy has cylindrical symmetry), or if there are vortices of any kind. We added the sentence “valid for systems with cylindrical symmetry” To the second line of section IV.

11) page 6, section VI: In Fig. 5, it is referred to the “normalized SC moment”. Can you specify the normalization (and indicate the shift by showing the “0” for each of the data sets)?

Reply> We corrected the caption to “Normalized magnetic moment $m/m(T\rightarrow 0)$ vs. temperature$. We think that there is no need to indicate the “0” since above Tc m=0 and it is clear from the figure.

12) page 7, paragraph in “The Knee”: Please clarify what is meant with “This might lead to two different Tc’s...” Can you indicate which theoretical work proposes different critical temperatures? The emergence of surface states for topological superconductors is by construction bound to the critical temperature of the bulk topological superconductor since the presence of the bulk topological state forces the surface state to exist. Please clarify or explain.

Reply> There is no theoretical work supporting two different Tc’s. It is our speculation. To make it clear we changed the sentence to “We speculate that this might lead to two different SC $T_c$s, one for the bulk and one for the surface.”

13) page 7, same paragraph. What is meant with “order parameters on different surfaces”? You refer to “Fermi surfaces” here? Then maybe “order parameters on different bands” might be more appropriate.

Reply> We changed to “different bands”.

14) Can you define the critical exponents with the appropriate behavior of rho and xi close to Tc? Instead of referring to “GL theory assumes a linear temperature of ...”, one could more generally state that the linear behavior of |\psi|^2 (if the latter is associated with the order parameter squared) is a result of the BCS (mean field) calculation where the critical exponent of the order parameter is 0.5.

Reply> we changed the sentence to “GL theory assumes, and BCS theory predicts, a linear temperature dependence of $\psi^2$. According to Eq.~\ref{eq:stiffness} this leads to the prediction that $n_{\rho}=1$.”

15) page 8, conclusion section: If understood correctly, one input for the data analysis is the simulation of the shape anisotropy for a given ring geometry to connect the measured signal to the magnetization of the material. However, this assumes that all volume of the ring is superconducting at T->0 and close to Tc where the important measurements are done. The latter might however be altered if the material is slightly inhomogeneous for example by slight changes of the impurity concentration which could easily generate a slightly smaller Tc in some regions such that the ring geometry of the superconducting material would be different (i.e. part of the ring is non-superconducting). Can the authors comment on this possibility and estimate/calculate in which direction the measured quantities would deviate from such an effect?

Reply> The purpose of the m(T) measurement in a field was to check how narrow is the transition. Although it is not an estimate, we can say that the transition is one of the narrowest we ever saw. Therefore, we were not concerned with the issue impurities. As for the effect of impurities on rho and xi measurement; impurities are expected to reduce the carrier density so reduce rho. Since lambda plays a rule in the calculation of xi, longer lambda will lead also to smaller xi. We did not include this discussion in the new manuscript.

Reviewer 2 Report

Conventional determination of superconducting stiffness ρs and coherence length ξ by measuring the penetration depth λ of a magnetic field and the upper critical field Hc2 in magnetic superconductors could lead to erroneous results since the internal field could be very different from the applied one. In this work, the authors measure DC superconducting properties in iron-based superconductor Fe1+ySexTe1−x with x 0.5 and y 0 in a zero-applied field to avoid contamination from magnetism making use of “Stiffnessometer” technique. The authors consider the method to be ideal for magnetic superconductors close to Tc, to measure both the penetration depth λ and coherence length ξ.  For Fe1+ySexTe1−x with x 0.5 and y 0 they find that λ and ξ are longer than previously reported and their temperature dependence agrees better with the Ginsburg-Landau predictions.

Despite several minor technical errors (see, e.g., page 4, Wang et al. citeWang20220.5) I recommend the paper for publication in Condensed Matter as it is.

Author Response

We thank review 2 for the report. The comment is:

Despite several minor technical errors (see, e.g., page 4, Wang et al. citeWang20220.5) I recommend the paper for publication in Condensed Matter as it is.

Reply> done

Reviewer 3 Report

In the present manuscript, the author developed a method to measure the penetration depth and coherence length of FeSeTe. Indeed, FeSeTe as an iron-based superconductor holds topological surface states, was studied intensively during the past few years. Overall, I think the experiments are well-designed. The theoretical model also agrees well with the experimental data. I believe this manuscript does meet the standard of Condensed Matter. However, I have some additional questions which should improve the clarity and consistency of the current manuscript.

Comment 1:In Figure 1, the picture of the sample doesn't look like FeSeTe single crystal at all. The author should explain in more detail about the sample preparation.

Comment 2: In the abstract, is the term "magnetic SC" commonly used? What is the point the author wants to highlight here?

Comment 3: Is there any new understanding can we get from Fig. 2a? Ferromagnetism is not new to this compound.

Author Response

We thank review 3 for the report. We address each comment individually.

Comment 1:In Figure 1, the picture of the sample doesn't look like FeSeTe single crystal at all. The author should explain in more detail about the sample preparation.

Reply> This sample has been cut by a femtosecond laser cutter. This process damages the surfaces and this is the reason the sample does not look like a single crystal. But, it looked like a single crystal before the cutting. We add a picture of the crystal before the cutting to figure 1 to show that.

Comment 2: In the abstract, is the term "magnetic SC" commonly used? What is the point the author wants to highlight here?

Reply> magnetic SC are materials that have a spontaneous magnetic field but also superconduct.

Comment 3: Is there any new understanding can we get from Fig. 2a? Ferromagnetism is not new to this compound.

Reply> We believe the referee meant Fig. 3a where we show hysteresis loop. Indeed, Ferromagnetism is not new to FST, but this figure is used to estimate the amount of free Fe. This is an important sample characterization which can be used to compare the quality of different samples.

Reviewer 4 Report

Peri et al have applied their novel technique for measuring superfluid density and critical fields to the material FeSe0.5Te0.5 (FST).  The technique is quite ingenious – fabricating a single-crystal sample into a ring-shaped geometry and applying a flux to the centre of the ring using a very narrow, normal-metal coil that is sufficiently long that the stray fields this coil creates are approximately uniform, so with very little pickup of the stray fields by a gradiometer-type pickup coil.  To preserve the flux state of the ring-shaped sample, a compensating flux is induced in the sample: the technique therefore provides a way of inducing a well-controlled supercurrent in the ring-shaped sample.  This in turn allows the superfluid density, critical currents and critical fields to be measured in a single experiment, and these are all important quantities for characterizing a given superconductor.  Very nicely, the experiment fits into the bore of a commercially produced SQUID magnetometer, allowing measurements to be carried out with high sensitivity, over a wide range of temperatures, and in an applied field.

As pointed out by the authors, their technique allows them to unambiguously measure superfluid density even when the sample is ferromagnetic, as is the case in FST.

The experiments have been carefully carried out and analyzed, with the analysis process clearly described.  The authors are very forthright about the various uncertainties and challenges faced by the experiment, which is much appreciated and will help with future interpretation of the results.  The discussion of the knee feature is very clear in this regard – the authors have carefully tested for reproducibility, and report the results as found.

This is a very good paper and publication in Condensed Matter is recommended after the following minor changes are made:

Page 1, paragraph 2: “cooper” -> “Cooper” 

Page 4, paragraph 1: resolve error in citation of Wang20220.5

Page 7, paragraph 2: “imprecation” -> “imperfection”

Bibliography: a few small typos due to BibTex formatting – e.g., chemical formulas (ReBe22) and capitalization of names (landau) – please check carefully.

Author Response

We thank review 4 for the very supportive report. We address each comment individually.

Page 1, paragraph 2: “cooper” -> “Cooper”

Reply> done.

Page 4, paragraph 1: resolve error in citation of Wang20220.5

Reply> done.

Page 7, paragraph 2: “imprecation” -> “imperfection”

Reply> done.

Bibliography: a few small typos due to BibTex formatting – e.g., chemical formulas (ReBe22) and capitalization of names (landau) – please check carefully.

Reply> done.

Reviewer 5 Report

The authors proposed a new method, called as stiffnessometer method. They determined the coherence length and the penetration depth by using this method together with the magnetic field penetration. The key point for this method is to use a long thin coil to pierce the superconducting ring, and then to measure the magnetic moment of the superconducting ring even the external magnetic field is zero, which leads to a direct measurement of the relationship between superfluid density rho_s and the wave vector A. The method seems interesting, and the derivations seem to be physically reasonable. However, I must say, the predictions are not explicit for the coherence and the penetration depth, in their methods these two key quantities need to be obtained by some sophiscated fitting process. Thus I conclude this new method, although interesting by its definition, is not better than other traditional methods, like magnetic field penetration, muSR, resistivity under magnetic fields, tunnel diode, STM etc. It is even worse that the method seems not effective at very low temperatures for the coherence length, for example see the data shown in Fig.4(b). The coherence length at low temperatures can be easily determined by the vortex core size through STM measurements. Both the penetration depth and the coherence length obtained by the proposed method seem longer than those reported by other methods. Especially for the penetration depth in the high temperature region, the determined value here is about 25-30 times longer than that determined by muSR (Fig.4(b)). In addition, they present some strange knees or kinks on the temperature dependence of penetration depth close to Tc (t~0.9), I wonder if this is induced by the method itself or due to basic features of the samples, since results from other methods (muSR) do not show that.

For these reasons, I cannot make a recommendation for publishing this work.

I would suggest the authors to use the method to measure the basic quantities of some superconductors with known values of coherence length and penetration depth, also serving as a comparison with the BCS formula. 

Author Response

We are sorry review 5 did not appreciate our manuscript. We address each comment individually.

The authors proposed a new method, called as stiffnessometer method. They determined the coherence length and the penetration depth by using this method together with the magnetic field penetration. The key point for this method is to use a long thin coil to pierce the superconducting ring, and then to measure the magnetic moment of the superconducting ring even the external magnetic field is zero, which leads to a direct measurement of the relationship between superfluid density rho_s and the wave vector A. The method seems interesting, and the derivations seem to be physically reasonable. However, I must say, the predictions are not explicit for the coherence and the penetration depth, in their methods these two key quantities need to be obtained by some sophiscated fitting process. Thus I conclude this new method, although interesting by its definition, is not better than other traditional methods, like magnetic field penetration, muSR, resistivity under magnetic fields, tunnel diode, STM etc. It is even worse that the method seems not effective at very low temperatures for the coherence length, for example see the data shown in Fig.4(b). The coherence length at low temperatures can be easily determined by the vortex core size through STM measurements. Both the penetration depth and the coherence length obtained by the proposed method seem longer than those reported by other methods. Especially for the penetration depth in the high temperature region, the determined value here is about 25-30 times longer than that determined by muSR (Fig.4(b)). In addition, they present some strange knees or kinks on the temperature dependence of penetration depth close to Tc (t~0.9), I wonder if this is induced by the method itself or due to basic features of the samples, since results from other methods (muSR) do not show that.

For these reasons, I cannot make a recommendation for publishing this work.

Reply> The referee completely missed the point of the paper. We claim that all the methods mentioned by the referee operate in magnetic field which couples to both the ferromagnet and the superconductor and hence the results are erroneous.

I would suggest the authors to use the method to measure the basic quantities of some superconductors with known values of coherence length and penetration depth, also serving as a comparison with the BCS formula.

Reply> Of course we measured superconductors with known values of stiffness and coherence length. See: A. Keren et al. Stiffness and coherence length measurements of ultra-thin superconductor, and implications to layered superconductors, Supercond. Sci. Technol. 35 075013 (2022).

Round 2

Reviewer 1 Report

The referee has examined the information at hand which consists in the revised manuscript and the replies to the referee comments from the previous round. As it turns out, there is no consensus about the recommendation for publication so far. While some referees seem convinced by the data, there are also some doubts about the reliability of the method compared to other established measurement methods. As the authors argue, their method yields very similar results for the penetration depth in known superconductors, but deviates for the present FeSeTe material because this one shows traces of magnetism and therefore other measurement methods are not reliable.

However, as also argued by referee 5, the present method relies on knowing the shape of the superconductor and subsequent modeling. Since especially for the FeSeTe system, the shape is not fully ideal, there might still be some additional deviations from the deduced penetration depth; a problem that was not there in the reference experiment.

An imprint of this error could already be seen from the T->0 value of the susceptibility which the authors estimate as between -1.3 and -1.15, a value beyond the theoretically expected value of -1 for ideal diamagnetism.

In summary, the referee nevertheless could recommend the publication of this work provided the authors add some words of caution to make the reader more aware about these issues.

Some more concrete comments on manuscript and reply:

Reviewer 1, comment 5, reply: There seems a misunderstanding here. Certainly, neither the magnetic moment is measured directly, nor the current since that cannot be done for a closed ring. The referee believes that simply the “VSM” mode is not explained clearly in the manuscript. Please spell out more clearly which quantities are measured and how one then calculates the magnetization from these, so it becomes more evident where systematic errors might propagate.

Reviewer 1, comment 11, reply: The referee respectfully disagrees with the authors in this point. Examining Fig. 5 (a), it turns out that for ring 2 and ring 3, there is visibly not a flat data at T>Tc. By putting the zero line for each dataset, it would be clear how much this deviates from zero, or whether there are even datapoints below zero.

Round 3

Reviewer 1 Report

The authors have implemented the proposed minor changes in the revised manuscript. Now, these points are more clearly presented in the manuscript such that the reader can clearly see the details of the data and understand the steps that need to enter the data analysis. The referee recommends acceptance of the manuscript for publication.